# Involvement of Mast-Cell-Tryptase- and Protease-Activated Receptor 2—Mediated Signaling and Urothelial Barrier Dysfunction with Reduced Uroplakin II Expression in Bladder Hyperactivity Induced by Chronic Bladder Ischemia in the Rat

**DOI:** 10.3390/ijms24043982

**Published:** 2023-02-16

**Authors:** Hidenori Akaihata, Kanako Matsuoka, Junya Hata, Yuki Harigane, Kei Yaginuma, Yu Endo, Hitomi Imai, Yuta Matsuoka, Akifumi Onagi, Ryo Tanji, Ruriko Honda-Takinami, Seiji Hoshi, Tomoyuki Koguchi, Yuichi Sato, Masao Kataoka, Motohide Uemura, Yasuhiko Igawa, Yoshiyuki Kojima

**Affiliations:** 1Department of Urology, Fukushima Medical University, 1 Hikarigaoka, Fukushima 960-1295, Japan; 2Department of Urology, Nagano Prefectural Shinshu Medical Center, 1332 Suzaka, Nagano 382-8577, Japan

**Keywords:** chronic bladder ischemia, bladder hyperactivity, urothelial barrier dysfunction, mast cell infiltration

## Abstract

We aimed to investigate the relationship between mast cell (MC) infiltration into the bladder with urothelial barrier dysfunction and bladder hyperactivity in a chronic bladder ischemia (CBI) rat model. We compared CBI rats (CBI group; n = 10) with normal rats (control group; n = 10). We measured the expression of mast cell tryptase (MCT) and protease-activated receptor 2 (PAR2), which are correlated with C fiber activation via MCT, and Uroplakins (UP Ia, Ib, II and III), which are critical to urothelial barrier function, via Western blotting. The effects of FSLLRY-NH2, a PAR2 antagonist, administered intravenously, on the bladder function of CBI rats were evaluated with a cystometrogram. In the CBI group, the MC number in the bladder was significantly greater (*p* = 0.03), and the expression of MCT (*p* = 0.02) and PAR2 (*p* = 0.02) was significantly increased compared to that of the control group. The 10 μg/kg FSLLRY-NH2 injection significantly increased the micturition interval of CBI rats (*p* = 0.03). The percentage of UP-II-positive cells on the urothelium with immunohistochemical staining was significantly lower in the CBI group than in the control group (*p* < 0.01). Chronic ischemia induces urothelial barrier dysfunction via impairing UP II, consequently inducing MC infiltration into the bladder wall and increased PAR2 expression. PAR2 activation by MCT may contribute to bladder hyperactivity.

## 1. Introduction

Lower urinary tract symptoms (LUTS), such as overactive bladder (OAB), greatly affect quality of life (QOL) in men and women, especially in the elderly [1,2]. The pathophysiology of lower urinary tract dysfunction (LUTD), which contributes to LUTS, is multifactorial [3]. It is well known that one cause of LUTD in both sexes is chronic bladder ischemia (CBI) induced by pelvic arterial occlusive disease, including atherosclerosis [4,5,6,7]. However, the mechanisms underlying the changes in lower urinary tract function induced by CBI have not been elucidated. 

On the other hand, recent attention has focused on the influence of mast cells (MCs) on LUTD. MCs not only directly mediate allergic or inflammation responses, but also sensitize sensory afferent nerve terminals in their proximity to sense potential damage [8]. A previous study revealed that MC infiltration into the bladder was observed in patients with detrusor overactivity (DO), which caused OAB [9]. It was also reported that MC infiltration into the bladder plays an important role in DO in rats with visceral hypersensitivity [10]. Mast cell tryptase (MCT), the most abundant MC protease, modulates neuronal activity by cleaving protease-activated receptor 2 (PAR2), which is expressed on C fibers. Protease-activated receptors (PARs) are a novel class of G-protein-coupled receptors cleaved by proteases, exposing tethered ligand domains that bind and activate the receptors [11]. PARs are important mediators of inflammation and repair [12]. PAR2 is the only member of the PAR family that can be activated by MCT [13]. MCT-mediated cleavage of sensory afferent nerve fiber-expressed PAR2 contributes to neurogenic inflammation and hyperalgesia in the intestine and joints of rodents [14,15]. The activation of sensory afferent-expressed PAR2 by MCT in the intestine was suggested to correlate with abdominal pain in patients with irritable bowel syndrome [13]. PAR2 has been shown to be expressed on bladder C fibers, which leads to bladder hyperactivity, and its overexpression was reported in a bladder with cyclophosphamide-induced cystitis [16]. MCs may cause bladder hyperactivity through the activation of increased PAR2 related to MCT. Recently, some studies reported that bladder inflammation including MC infiltration, which might increase PAR2 expression, was affected by the change in Uroplakin (UP) [17,18]. UP Ia, UP Ib, UP II and UP III, which are transmembrane proteins in the apical membrane of the bladder urothelial cells, act as a urothelial exceptional barrier to water, small non-electrolytes and protons [19]. UPs are synthesized in large quantities only by terminally differentiated urothelial cells. UPs play an important role in the homeostasis of the urinary bladder mucosa during contraction and relaxation [20,21]. When the permeability barrier consisting of UPs is breached, irritants in the urine reach the bladder submucosa structure and musculature [22,23]. Female rats infused with protamine sulphate intravesically showed a patchy distribution of UPs and infiltration of inflammatory cells including MCs in the bladder [17]. The cystitis rat model, which is an interstitial cystitis model and autoimmune to UP2, showed loss of urothelial barrier function and MC infiltration [24]. The urothelial barrier dysfunction correlated with UPs may contribute to MC infiltration in the bladder. Although MC infiltrations, increased PAR2-associated activation and the impaired UPs may affect lower urinary tract function, whether these changes contribute to LUTD caused by chronic ischemia has not been established.

In the present study, we aimed to investigate possible changes in UPs, MCT and PAR2 expression and their relationship with bladder hyperactivity in a rat model of CBI [25], and discussed the mechanisms underlying LUTD caused by chronic ischemia.

## 2. Results

### 2.1. Arterial-Occlusion-Caused Bladder Hyperactivity Correlated with Chronic Ischemia

Metabolic cage studies showed that mean and maximum void volumes were significantly smaller in the CBI group than in the control group (Table 1). Elastica–Masson (EM) stains demonstrated that the mean arterial wall thickness was significantly greater in the CBI group than in the control group (control vs. CBI, 82.0 ± 26.3 μm vs. 116.3 ± 23.8 μm. *p* < 0.01) (Figure 1A). Western blot analysis showed the expression of HIF1α in the bladder was significantly increased in the CBI group as compared to the control group (*p* = 0.03) (Figure 1B).

### 2.2. Increased Mast Cell Infiltration and PAR2 Expression in Rats with CBI

The methylene blue staining showed the number of mast cells in the bladder suburothelial layer was significantly greater in the CBI group than in the control group (control vs. CBI, 0.8 ± 0.6 numbers/area vs. 1.5 ± 0.7 numbers/area, *p* = 0.03) (Figure 2A). The immunohistochemical staining of PAR2 demonstrated that the number of PAR2-positive cells in the bladder was significantly greater in the CBI group than in the control group (control vs. CBI, 34.3 ± 5.5 numbers/area vs. 68.9 ± 24.7 numbers/area, *p* < 0.01) (Figure 2B). The Western blot analysis revealed the expression of MCT (*p* = 0.02) and PAR2 (*p* = 0.02) in the rat bladders was significantly greater in the CBI group than in the control (Figure 2C).

### 2.3. Effects of the PAR2 Antagonist on Bladder Function in CBI Rats

The effects of FSLLRY-NH2, a PAR2 antagonist, administered intravenously on bladder function in the CBI rats were evaluated by conscious cystometrogram. The cystometrograms were recorded continuously after intravenous injection of saline, and cumulative dosages of 10, 30 and 100 µg/kg of FSLLRY-NH2 in sequence. As shown in Figure 3A, micturition intervals and void volumes were increased after an injection of 10 µg/kg FSLLRY-NH2. With the 10 μg/kg FSLLRY-NH2 injection, the micturition interval (*p* = 0.03) was significantly longer, and the bladder capacity (*p* = 0.03) and void volume (*p* = 0.03) were significantly larger, as compared to those with a saline injection. There were no significant differences in residual volume (*p* = 1.00), baseline pressure (*p* = 1.00), threshold pressure (*p* = 0.74), maximum pressure (*p* = 0.37) or bladder compliance (*p* = 1.00) between saline and 10 μg/kg FSLLRY-NH2 injections (Figure 3B). With the injection of either 30 or 100 μg/kg FSLLRY-NH2, no cystometric parameters were significantly different from those with the saline injection.

### 2.4. Reduced Uroplakin II Expression in the Bladder Urothelial Cells in CBI Rats

With immunohistochemical staining, UP-Ia-, UP-Ib-, UP-II- and UP-III-positive cells were located mostly on the urothelium. The percentage of UP-II-positive cells was significantly lower in the CBI group than in the control group (control vs. CBI, 92.1 ± 0.1% vs. 63.0 ± 0.1%, *p* < 0.01). No differences in the percentage of UP-Ia-, UP-Ib- or UP-III-positive cells were evident between the control group and CBI group (control vs. CBI, UP Ia: 94.9 ± 5.3% vs. 94.2 ± 4.0%, *p* = 0.55; UP Ib: 54.4 ± 5.0% vs. 54.3 ± 4.1%, *p* = 0.96; UP III: 47.5 ± 5.8% vs. 49.0 ± 4.9%, *p* = 1.00) (Figure 4A). The Western blot analysis using rat bladders also showed the expression of UP II was significantly decreased in the CBI group as compared to the control group (*p* = 0.01). No significant differences were found in UP Ia (*p* = 0.42), UP Ib (*p* = 1.00) or UP III (*p* = 0.51) expression in the bladders between the two groups (Figure 4B).

## 3. Discussion

In this study, we confirmed previous observations that pelvic arterial occlusion in rats causes ischemia-reperfusion injury and bladder hyperactivity, as indicated by a decreased bladder capacity and void volumes [25]. Our results show increased MC numbers and MCT expression in the rat bladders with CBI. Previous studies suggested that increased MC–nerve interaction might be a cause of bladder hyperactivity [10,26]. Keith and Saban [26] reported an intimate anatomic relationship between MCs and primary afferent nerve fibers in the guinea pig bladder and concluded that the imbalance in this mutual exchange could potentially cause bladder hyperactivity. The afferent nerve activation related to MCs and MCT may cause bladder hyperactivity in rats.

MCs have been demonstrated to stimulate afferent nerves through the activation of PAR2 by MCT [13]. MCT cleaves PAR2 at a specific site and exposes the tethered ligand domain, resulting in PAR2 activation [27]. It is believed that overactivation of bladder afferent nerve fibers is involved in the initiation of OAB symptoms and detrusor overactivity [28]. In our study, the number of PAR2-positive cells and the expression of PAR2 were increased in the bladders of the CBI rats as compared to those of the control rats. These results suggest that the increased activation of PAR2 on afferent nerve fibers by MCT from MCs contribute to bladder hyperactivity associated with chronic ischemia. In this study, intravenous administration of 10 μg/kg FSLLRY-NH2 increased the micturition interval, bladder capacity and void volume in the CBI rat model. This role of FSLLRY-NH2 in ameliorating bladder hyperactivity also indicated that the activation of PAR2 on afferent nerve fibers was involved in the development of bladder hyperactivity associated with chronic ischemia. However, neither 30 nor 100 µg/kg FSLLRY-NH2 intravenous injections significantly affected any cystometric parameters in the CBI rats. In this non-linear relationship, the effects of increasing dosages of a given compound appear to increase to a maximum and then decrease, a phenomenon known as the inverted U-shaped dose–effect curve. The inverted U-shaped dose–effect curve has been reported for many active compounds. However, the mechanisms responsible for the inverted U-shaped dose–effect curve are poorly understood. It has been suggested that compounds with multiple interactions occurring among several receptors might have an inverted U-shaped dose–effect curve due to the activation of several pathways and that this curve is initiated by a defense mechanism activated in response to excessive high concentrations of a given active compound [29,30,31]. PAR2 has been shown to modulate a variety of physiological processes, including vasodilation, neurogenic inflammation and smooth muscle tone [16,32]. Only appropriate FSLLRY-NH2 dosage may improve bladder hyperactivity caused by chronic ischemia. PAR2 has been reported to be expressed on afferent C fibers. The afferent pathways innervating the urinary bladder consist of myelinated A-delta fibers and unmyelinated C fibers. The hyperexcitability of C fibers in bladder afferent pathways is suggested to be responsible for bladder hyperactivity [33]. Dattilio et al. reported up-regulation of PAR2 expression in the bladder of rats with cyclophosphamide-induced cystitis. They suggested that the activation of PAR2 on bladder C fibers may exacerbate focal inflammation to a more global dysfunction by activating lower urinary tract neural pathways [16]. In the same way, PAR2 on bladder C fibers might be activated by MCT in chronically ischemic bladders, resulting in bladder hyperactivity.

The CBI group showed decreased expression of UP II in the bladder and a lower percentage of UP-II-positive cells in the urothelium as compared to the control group. These results suggest chronic ischemia impaired normal urothelial barrier function through reduced expression of UP II in the rat. Several studies have reported an association between urothelial barrier dysfunction and inflammation-related LUTD. Spontaneously hypertensive rats (SHRs), which is a rat model of OAB, showed bladder inflammation and impaired urothelial barrier function with a decreased number of UPs [18]. In the SHRs, supporting urothelial barrier function improved frequent urination by reducing inflammation [18]. Previous studies showed increased inflammatory cells and decreased UPs in OAB patients compared to healthy individuals [34,35]. Associations between urothelial barrier dysfunction and increased MCs in the bladder were also reported. Cetinel et al. reported an increased number of MCs in rat bladders with urothelial barrier dysfunction caused by protamine sulphate. They also demonstrated that protecting urothelial barrier function prevented MC infiltration in rat bladders [17]. Our results suggest chronic-ischemia-induced MC infiltration in the bladder occurs when urothelial barrier function is impaired. In this study, only UP II was impaired in the CBI rats. Among UPs, UP II or UP III deficiency is often suggested to contribute to LUTD. Aboushwareb et al. reported that both UP II knockout (KO) mice and UP IIIa KO mice revealed demonstrable non-voiding contractions [36]. In mice, immunization with UP II induced considerable bladder inflammation and decreased urine output per voiding without chronic pain [24]. Altogether, we can speculate that chronic ischemia induces urothelial barrier dysfunction by down-regulation of UP II, resulting in bladder inflammation, which causes MC infiltration and increased PAR2 expression in the bladder.

Several limitations to this study must be considered when interpreting the results. First, we did not measure actual bladder blood flow. However, we believed that increased HIF-1α expression in the bladder and iliac arterial wall thickening reflected chronic ischemia [37,38]. Second, we have not proven directly that PAR2 activation causes bladder hyperactivity in the CBI rat model. However, we believe increased PAR2-positive cells and expression of PAR2 in the bladder of the CBI rat model and the role of the PAR2 antagonist in ameliorating bladder hyperactivity indicate that PAR2 activation contributes to bladder hyperactivity caused by chronic ischemia in rats. Third, we did not analyze the expression of C fibers or nerve markers or establish a co-localization with PAR2. However, we believe that the effect of the PAR2 antagonist on the bladder function of the CBI rat model indicates an association between PAR2 and C fibers. Fourth, this study did not evaluate transient receptor potential A1 (TRPA1) sensitization in the bladder. Some studies reported that the PAR2-dependent mechanism sensitizes TRPA1 to induce hypersensitivity in C fibers. PAR2 activation might also cause TRPA1 sensitization in these CBI rats, which should be clarified by a future study. Fifth, we did not confirm down-regulation of UP II actually leads to barrier dysfunction. However, we consider MC infiltration to indicate urothelial barrier dysfunction.

In conclusion, our results suggest that chronic bladder ischemia induces urothelial barrier dysfunction associated with reduced expression of UP II, and consequently MC infiltration and increased MCT from MCs and then PAR2 activation, which may contribute to LUTD, including bladder hyperactivity, through the facilitation of bladder afferent—likely including C fibers—activities.

## 4. Material and Methods

This experimental protocol complied with set guidelines for animal experiments. This protocol was reviewed and approved by the Animal Ethics Committee of Fukushima Medical University (#2019015).

### 4.1. Experimental Design

Adult male Sprague Dawley rats (16 weeks old) were divided into two groups (control and CBI; n = 10 each). The CBI group underwent balloon endothelial injury of bilateral iliac arteries and received a 2% cholesterol diet for 8 weeks to induce arterial occlusive disease-related chronic ischemia. The control group received a regular diet for 8 weeks. After monitoring urine output for 24 h, the bladders and common iliac arteries were harvested for molecular biological and histological examinations. We used Western blotting to measure expressions of UP Ia, UP Ib, UP II, UP III, MCT, PAR2 and HIF1α (an oxidative stress marker) in the bladder. Bladders were processed for immunohistochemical staining and methylene blue staining. The effects of FSLLRY-NH2, a PAR2 antagonist, administered intravenously on the bladder function of CBI rat models were also evaluated via conscious cystometrogram.

### 4.2. Iliac Arterial Endothelial Injury

The procedure for creating iliac artery endothelial injury has been described previously [37,39]. Under anesthesia with 2% isoflurane, 2-Fr Fogarty arterial embolectomy catheters were passed through the femoral arteries into the common iliac arteries of rats. The balloons were inflated with air, then withdrawn from the common iliac artery to the femoral artery ten times on each side to induce arterial endothelial injury in the external and common iliac artery.

### 4.3. Metabolic Cage Studies

Then, 8 weeks after the Iliac arterial endothelial injury, each rat was placed in an individual metabolic cage (CT-10S; CREA) for 24 h over 3 consecutive days with a 12/12 h dark/light cycle. An electric balance with a urine collection system was placed under the metabolic cage and connected to the MacLab 4/20 data acquisition board. Food and water were supplied ad libitum. The mean void volume (mL) and maximum void volume (mL) were evaluated.

### 4.4. Histological Examination

The common iliac arteries and urinary bladders from each group were fixed in 10% neutral-buffered formalin, embedded in paraffin and cut into 5 μm sections. Slides were cleared with xylene, dehydrated and used for staining with Elastica–Masson (EM) stains. The common iliac arterial wall thickness was determined by averaging the wall thickness at four distinct locations in each sample using a microscope and cell Sens Dimension image analysis software (OLYMPUS, Tokyo, Japan) [37]. The bladder was processed for methylene blue staining and immunohistochemical staining. The number of MCs was counted in 5 consecutive fields (×400), and the mean values were used in the subsequent statistical analysis [10]. For immunohistochemical examinations of bladder tissue, sections were deparaffinized, and endogenous peroxidase was quenched with 0.3% H_2_O_2_. Nonspecific immunoglobulin G binding was blocked with 5% skim milk. Sections were incubated overnight at 4 °C with primary antibodies for anti-RAR2 (GR 324700-11, 1:200; Abcam, Cambridge, UK), anti-UP Ia (GR160579-1, 1:200; Abcam), anti-UP Ib (GR153380-5, 1:200; Abcam), anti-UP II (14277, 1:200; Proteintech, Rosemont, IL, USA) and anti-UP III (GR31809451, 1:200; Abcam). After rinsing three times in phosphate-buffered saline for 5 min, appropriate species-directed secondary antibodies (Signal Stain Boast IHC Detection Reagent; Cell Signaling Technology, Danvers, MA, USA) were applied to the sections at room temperature. When substances staining positive were located in the cytoplasm as dark-brown granules, these cells were judged as positive in the immunohistochemical evaluation. The percentage of UP-positive cells on the urothelium was assessed. The percentage of UP-positive cells in each rat was calculated in four randomly selected high-power fields (×400).

### 4.5. Protein Extraction, SDS-PAGE and Western Blotting

The whole bladders were frozen with liquid nitrogen. The tissues were homogenized, and proteins were extracted with urea (8 M) and dithiothreitol (10 mM). Total protein concentrations of samples were measured with a BCA Protein Assay Kit (Thermo Fisher Scientific Inc., Waltham, MA, USA). Samples were then mixed with 5× sodium dodecyl sulphate (SDS) sample buffer and boiled (4 min). Each sample, containing 10 μg total protein, was loaded onto an acrylamide gel, and proteins were separated by electrophoresis and blotted onto a PVDF membrane. The membranes were blocked for 1 h with 1% polyvinylpyrrolidone in TBS-T buffer (20 mM Tris pH 7.5, 0.5 M NaCl, 0.1% Tween 20). After washing with TBS-T, the membranes were kept at 4 ºC overnight with the primary antibody (1:1000 dilution) including the rabbit monoclonal antibodies anti-HIF1α (#I1212, Santa Cruz, Dallas, TX, USA), anti-MCT (GR323598-3, Abcam), anti-RAR2 (GR 324700-11, Abcam), anti-UP Ia (GR160579-1, Abcam), anti-UP Ib (GR153380-5, Abcam), anti-UP II (14277, Proteintech), anti-UP III (GR31809451, Abcam) and the mouse monoclonal antibody anti-β actin (#90422, SIGMA). The membranes were then washed four times with TBS-T and incubated for 50 min with horseradish peroxidase-conjugated second antibodies (1:1000 dilution; Promega KK, Madison, WI, USA). All antibodies were diluted with Can Get Signal Immunostain (Toyobo Company, Osaka, Japan). Following three washings with TBS-T, bands were visualized using SuperSignal West Dura Extended Duration Substrate (Thermo Fisher Scientific Inc., Waltham, MA, USA) and imaged using a ChemiDoc XRS plus system (BIO-RAD, Hercules, CA, USA). We washed the membranes with WB Stripping Solution Strong, which allowed us to use the same membranes twice (Nacalai Tesque, Kyoto, Japan). For the first wash, anti-HIF1α, anti-MCT, anti-RAR2, anti-UP Ia, anti-UP Ib, anti-UP II or anti-UP III was used. For the second wash, only anti-β actin was used.

### 4.6. Cystometry in Conscious, Freely Moving Rats

For the intravenous administration of FSLLRY-NH2 (#63203, SIGMA), a PAR2 antagonist, six CBI rats, which were not used for other experiments in this study, were subjected to chronic catheterization of the right jugular vein. Under general anesthesia with isoflurane, a 1 cm longitudinal incision was made on the ventral surface of the neck, 1 cm right from the trachea. The right jugular vein was exposed and freed from its surrounding connective tissue. A 10 cm long silicone catheter (Becton, Dickinson and Company, New Jersey, USA) filled with heparinized (50 U/mL) saline was inserted into the superior vena cava through the jugular vein and secured in place with ligatures. The free end of the catheter was tunneled under the skin and exteriorized at the nape. After surgery, catheters were flushed with heparinized saline daily. For cystometric studies, a polyethylene catheter was previously implanted into the bladder [37,38]. Briefly, the bladder dome was delivered outside the rat’s body, and a small incision was made under anesthesia with 2% isoflurane. A polyethylene catheter (Nihon Bioresearch Inc., Aichi, Japan) with a cuff was inserted through the incision and anchored. The proximal end of the catheter was tunneled subcutaneously as an exit at the nape of the neck. Cystometric studies were performed in conscious, freely moving rats 3 days after the catheter implantation. The bladder catheter was connected to a pressure transducer (Becton, Dickinson and Company, Franklin Lakes, NJ, USA) and microinjection pump (Nihon Kohden, Tokyo, Japan). Room-temperature saline was instilled into the bladder at a rate of 10 mL/h. The pressure transducer was connected to an AP-601G transducer amplifier (Nihon Kohden, Tokyo, Japan) and subsequently connected to a MacLab 4/20 data acquisition board. An electric balance with a urine collection system was placed under the metabolic cage and connected to the MacLab 4/20 data acquisition board. Rats were given at least 30 min for voiding patterns to stabilize. The cystometrograms were recorded continuously after intravenous injection of saline, and 10 µg/kg, 20 µg/kg (cumulative dosage 30 µg/kg) and 70 µg/kg (cumulative dosage 100 µg/kg) FSLLRY-NH2 doses in sequence. The volume injected was 0.25 mL, and the duration of injections was 1 min. At least 45 min was allowed to pass between injections. When the rats voided after 45 min from saline or drug injection, the next dose of drug was injected thereafter. Cystometric parameters were evaluated for the interval between the voiding just after saline or drug injection and the voiding before the next dose of drug injection. The cystometric parameters investigated were the micturition interval (time period between two maximum voiding pressures), bladder capacity (infused volume/the number of micturitions), void volume (void volume per micturition resisted registered in a urine collection cup), residual volume (bladder capacity—void volume), basal pressure (minimum pressure between two micturitions), threshold pressure (intravesical pressure immediately before micturition), maximum pressure (the maximum bladder pressure during a micturition cycle) and bladder compliance (bladder capacity/threshold pressure—baseline pressure).

The ratio (%) of each cystometric parameter with each dose of FSLLRY-NH2 injections relative to the corresponding parameter with saline was used for evaluation of the drug’s effect.

### 4.7. Statistical Analysis

Data were analyzed using the IBM SPSS Statistics 21 software. Groups were compared using the Mann–Whitney U Test. The Friedman and Bonferroni tests were used to evaluate the dose-dependent effects of the FSLLRY-NH2 on the cystometric parameters in CBI rats. Values of *p* < 0.05 were considered significant.

## Figures and Tables

**Figure 1 ijms-24-03982-f001:**
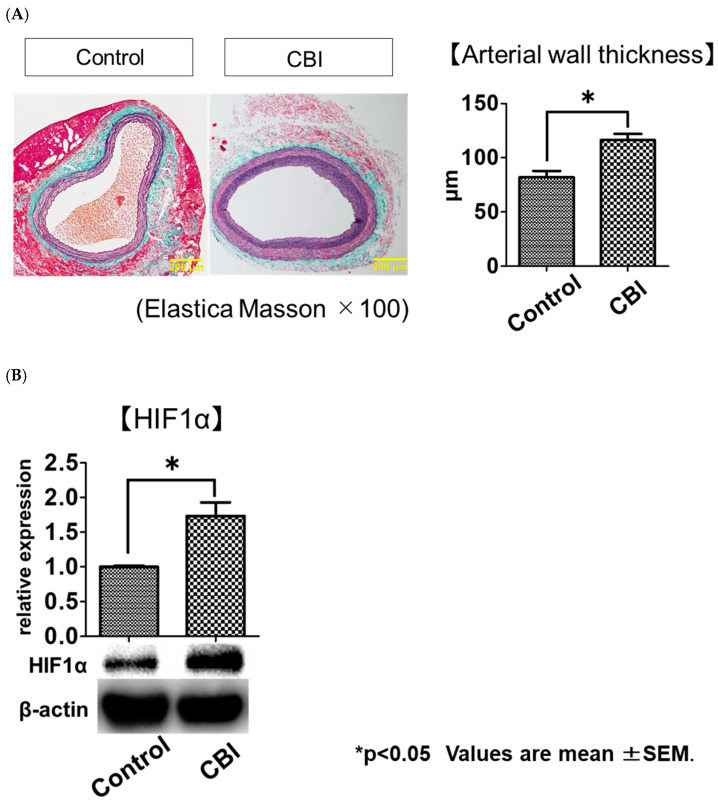
(**A**) Elastica–Masson staining of the same cross-sections of common iliac artery from rats in the control (10 rats, 20 vessels) and CBI (10 rats, 20 vessels) groups (100× magnification; scale bars, 200 μm). Bar graphs show average wall thickness of the common iliac artery in control and CBI groups. Values represent mean ± SEM. * *p* < 0.05 (Mann–Whitney U Test). (**B**) Expression of HIF-1α, an oxidative stress marker, at the protein level in the bladder as assessed using Western blotting. The bar represents the mean ± SEM of 6–9 determinations, each from a different bladder. * *p* < 0.05 (Mann–Whitney U Test).

**Figure 2 ijms-24-03982-f002:**
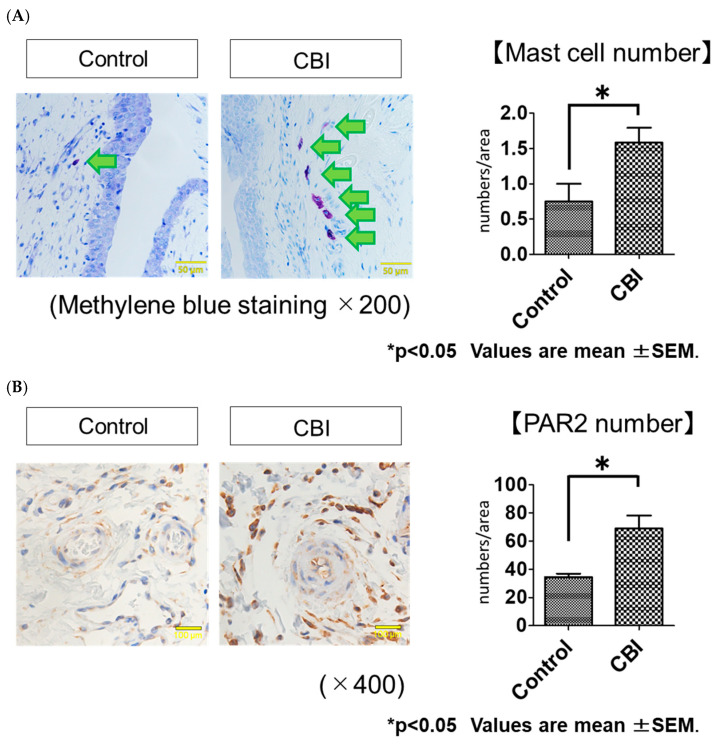
(**A**) Mast cell infiltration in the suburothelial layer of the bladders from the control and CBI groups as seen with methylene blue staining (200× magnification; scale bars 50 μm). The arrows point to mast cells in the bladder suburothelial layer. Bar graphs show the numbers/area of mast cells in the control and CBI groups. * *p* < 0.05 (Mann–Whitney U Test). (**B**) Immunohistochemical examinations of PAR2 in the suburothelial layer of the bladder specimens from the control and CBI groups (400× magnification; scale bars 100 μm). Bar graphs show the numbers/area of PAR2-positive cells in the control and CBI groups. Values represent mean ± SEM. * *p* < 0.05 (Mann–Whitney U Test). (**C**) Expressions of MCT and PAR2 at the protein level in the bladder as assessed using Western blotting. The bar represents the mean ± SEM of 6–9 determinations, each from a different bladder. * *p* < 0.05 (Mann–Whitney U Test).

**Figure 3 ijms-24-03982-f003:**
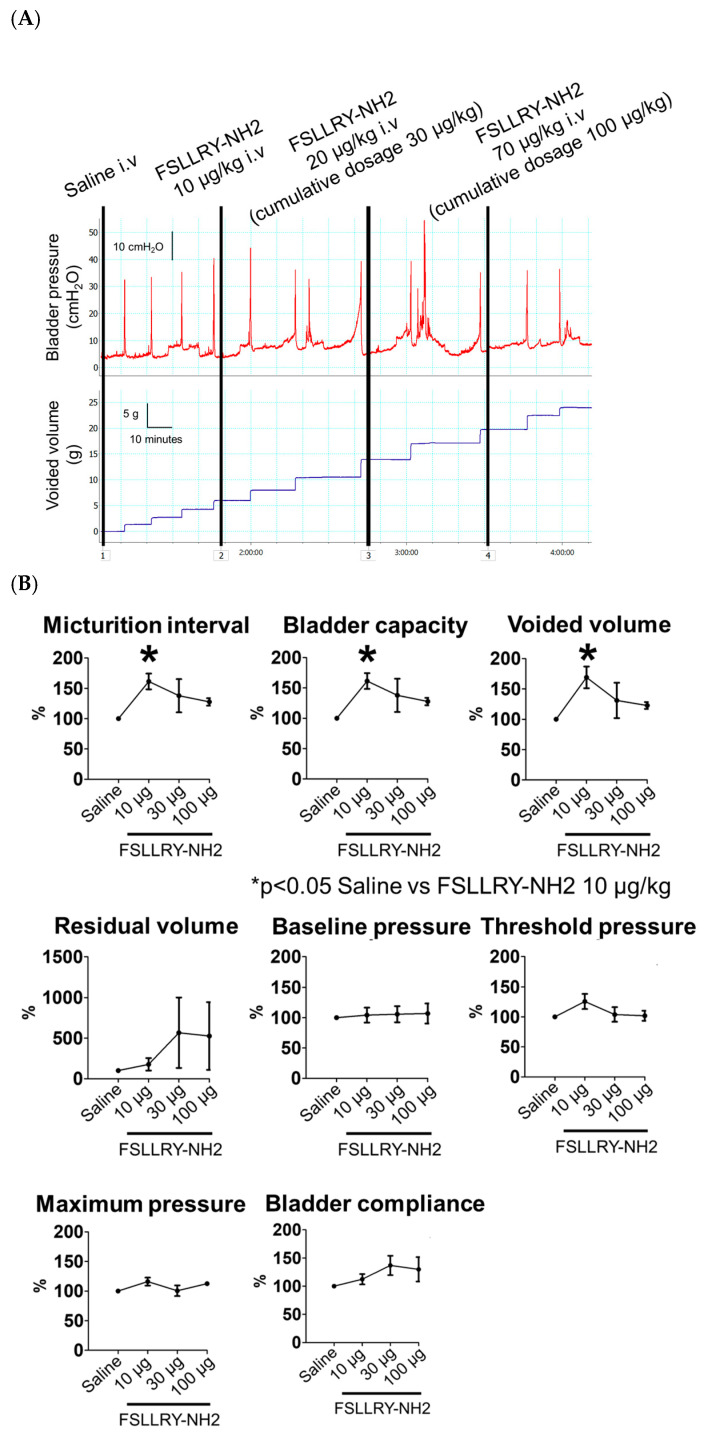
(**A**) Representative cystometrogram recordings with intravenous administrations of FSLLRY-NH2, a PAR2 antagonist, in a rat with chronic bladder ischemia (CBI). The cystometrograms were recorded continuously after intravenous injection of saline, and 10 µg/kg, 20 µg/kg (cumulative dosage 30 µg/kg) and 70 µg/kg (cumulative dosage 100 µg/kg) of FSLLRY-NH2 in sequence. Scale bars represent 10 min, 10 cm H_2_O and 5 mL. (**B**) Effects of FSLLRY-NH2, a PAR2 antagonist, on cystometric parameters of the rat model of chronic bladder ischemia (CBI). Values (mean ± SEM) are expressed as the ratio (%) of cystometric parameters with each dose of FSLLRY-NH2 injection relative to the corresponding parameters with saline. * *p* < 0.05 (Friedman and Bonferroni tests).

**Figure 4 ijms-24-03982-f004:**
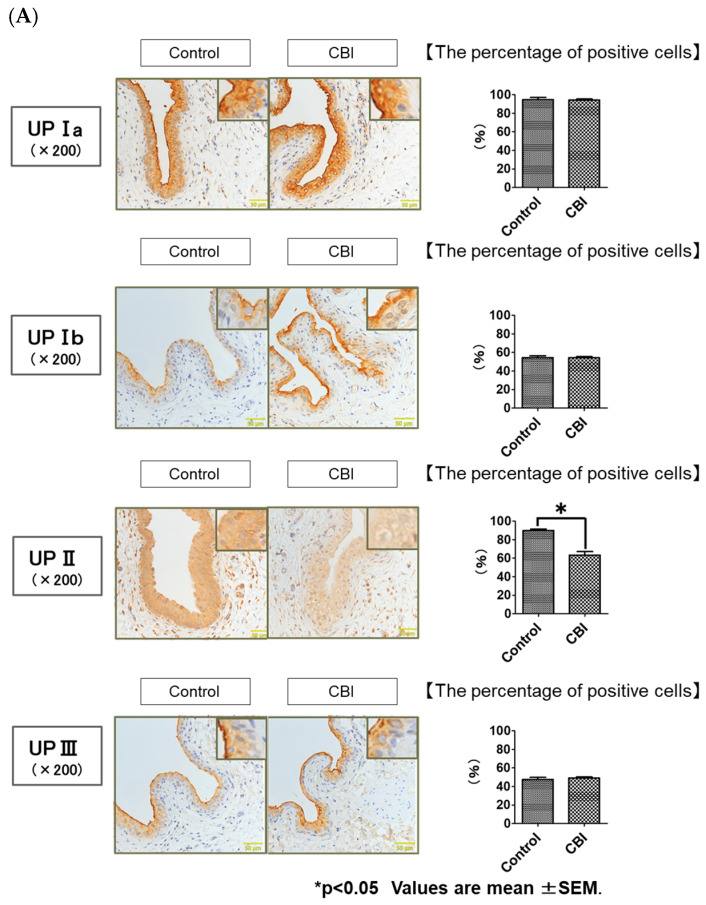
(**A**) Immunohistochemical examinations of the bladder specimens from the control and CBI groups (200× magnification; scale bars 50 μm). Bar graphs show the average percentage of UP-positive cells in the control and CBI groups. Values represent mean ± SEM. * *p* < 0.05 (Mann–Whitney U Test). (**B**) Expression of Uroplakins (UPs) at the protein level in the bladder according to Western blot. Each bar represents the mean ± SEM of 6–9 determinations, each from a different bladder. * *p* < 0.05 (Mann–Whitney U Test).

**Table 1 ijms-24-03982-t001:** Metabolic cage studies in the control and CBI groups.

	Control (n = 10)	CBI (n = 10)	*p* Value
Mean void volume (mL)	1.46 ± 0.33	1.01 ± 0.21	*p* < 0.01
Maximum void volume (mL)	2.62 ± 0.60	2.01 ± 0.41	*p* = 0.02

Values represent mean ± standard deviation.

## Data Availability

Not applicable.

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
