# Peer review of "Involvement of Mast-Cell-Tryptase- and Protease-Activated Receptor 2—Mediated Signaling and Urothelial Barrier Dysfunction with Reduced Uroplakin II Expression in Bladder Hyperactivity Induced by Chronic Bladder Ischemia in the Rat"

_ijms, 2023, doi:10.3390/ijms24043982_

Round 1

Reviewer 1 Report

In the study by Akaihata and coworkers, the investigators used a model of chronic bladder ischemia (CBI) to investigate the relation between mast cell infiltration, alterations in the mucosal permeability barrier and bladder hyperactivity. CBI was induced by injury to the iliac arteries and bladder function assessed 8 weeks after injury. During this time, animals were fed with a 2% cholesterol diet. At this time point, bladder function was evaluated and bladders collected for tissue analysis. In samples from CBI bladders, the authors observed an upregulation in the number of mast cells, accompanied by increased levels of mast cell tryptase and PAR2. They also found changes in the expression of uroplakin II, consistent with damage to the most superficial urothelial layer. Finally, treatment with a PAR2 inhibitor improved bladder function in CBI animals.

Overall, an interesting study but it needs to be improved before acceptance for publication:

- - The first point I would like to address refers to the Introduction. Although I acknowledge the relevance of the study, the link between CBI, mast cells, PAR2 and uroplakins needs to be made clearer. Does PAR2 affect uroplakins?

- - The introduction is somewhat disjunct and, while there is evidence for each factor contributing to bladder dysfunction, the hypothesis they would be operating in a sequential manner needs to be better explained and supported.

-          For western blot analysis, it is important to refer if samples are from the whole bladder or if there has been separation of the mucosa.

-          As the authors indicate that MCT act on PAR2, is it possible to perform immunohistochemical analysis to demonstrate co-localization or proximity? Also, why is PAR2 upregulated? Also, the authors show that PAR2-positive cells are increased but, as they refer in the Discussion that PAR2 may also be present in C fibres, is it possible to analyse the expression of C-fibres markers and establish a co-localization with PAR2.

-          While bladder function in CBI animals was certainly affected by the PAR2 antagonist, it is not clear how different these urodynamic records are from normal animals of if the compound has some deleterious effects on intact animals. Not addressed here, but it would be interesting to discuss what would happen if this model would be replicated in aged animals, in analogy to elderly patients who often present vascular alterations.

-          In the urodynamic studies with PAR2 inhibitors, the effects are cumulative and that should be stressed. For example, the effects of 100 µg represent, in fact, the effects of 140 µg as dosages were injected in the same animal.

-          A critical point relies in the lack of tissue analysis after PAR2 inhibitor. Why was it not done? One could expect some effects. Also, why did authors choose this inhibitor? Are there any MCT inhibitors available that could be used to validate the results obtained?

-          Also important, in discussion, the authors refer to “number of PAR2 positive cells” but some sentences ahead refer the presence of PAR2 in afferent fibres. In figure 2, it seems there are no fibres stained, only cells. Which cells are these? Is there any co-localization with neuronal markers (such as beta III tubulin)? Where are these cells located? Lamina propria?

-          The 1st paragraph of the discussion should be divided into sections. There are too many subjects being discussed at the same time. The hypothesis of the lack of effect of 30 and 100 µg needs to be better explained. Are there any alternative hypotheses?

Minor points:

-          In section 2.1, there is no need to indicate values are they are described in table 1.

-          In figure 3B, please include some notification (asterisk) showing statistical differences. As in other figures, include that information in the legend, as well.

-          To the less experienced reader, in the discussion it is advisable to clarify that the myelinated A- fibres are delta fibres (and not alfa or beta).

Author Response

Thank you for Reviewers comment. We hope we have made our paper understandable.

Point by point response

Reply to Reviewer 1

The first point I would like to address refers to the Introduction. Although I acknowledge the relevance of the study, the link between CBI, mast cells, PAR2 and uroplakins needs to be made clearer. Does PAR2 affect uroplakins?

The introduction is somewhat disjunct and, while there is evidence for each factor contributing to bladder dysfunction, the hypothesis they would be operating in a sequential manner needs to be better explained and supported.

# Response

Thank you very much for this valuable comment. We consider that chronic ischemia causes first urothelial barrier dysfunction with down-regulation of UP II, which allows irritants in the urine to reach the bladder suburothelial structure. The irritants induced inflammation with increased mast cells in the bladder. Consequently, mast cell tryptase released by mast cells activates the nerves with PAR2 expression, resulting in bladder hyperactivity. We have modified the introduction and discussion sections by adding these sequential hypothesis.

For western blot analysis, it is important to refer if samples are from the whole bladder or if there has been separation of the mucosa.

# Response

In this study, we used the whole bladder for western blot analysis. The change has been carried out in the method section. Please see page 12 line 323.

As the authors indicate that MCT act on PAR2, is it possible to perform immunohistochemical analysis to demonstrate co-localization or proximity? Also, why is PAR2 upregulated? Also, the authors show that PAR2-positive cells are increased but, as they refer in the Discussion that PAR2 may also be present in C fibres, is it possible to analyse the expression of C-fibres markers and establish a co-localization with PAR2.

Also important, in discussion, the authors refer to “number of PAR2 positive cells” but some sentences ahead refer the presence of PAR2 in afferent fibres. In figure 2, it seems there are no fibres stained, only cells. Which cells are these? Is there any co-localization with neuronal markers (such as beta III tubulin)? Where are these cells located? Lamina propria?

# Response

Thank you very much for your suggestions.

In Figure 2B, we evaluated the number of PAR2-positive cells in the suburothelial layer of the bladder, which is now clearly stated in the legends of the Figure 2 (page 5 line 120).

A previous study demonstrated upregulation of PAR2 expression in the bladder, as well as colocalization of PAR2 and TRPV1 in bladder C-fibres (Dattilio A, J Urol. 2005). However, we have not determined which cells are these PAR2-positive cells. Regretfully, we do not have enough time to perform further immunohistochemical analysis suggested by the reviewer. We considered that this is a limitation of our study.

We have now stated this limitation (page 11 line 255-258).  

While bladder function in CBI animals was certainly affected by the PAR2 antagonist, it is not clear how different these urodynamic records are from normal animals of if the compound has some deleterious effects on intact animals. Not addressed here, but it would be interesting to discuss what would happen if this model would be replicated in aged animals, in analogy to elderly patients who often present vascular alterations.

# Response

In this study, we did not evaluate the effect of the PAR2 antagonist on bladder function of intact rats. We guess that the changes in bladder function in intact rats by PAR2 antagonist are not expected to be as great as in the CBI rat model because intact rats have lower PAR2 expression in the bladders as compared with CBI rats. Thank you very much for your suggestion about aged animals. But, it would be beyond the aims of the present study. We will try it next.

In the urodynamic studies with PAR2 inhibitors, the effects are cumulative and that should be stressed. For example, the effects of 100 µg represent, in fact, the effects of 140 µg as dosages were injected in the same animal.

# Response

We described cumulative dosages in the original manuscript. In fact, we injected 10 µg/kg, 20 µg/kg (cumulative dosage 30 µg/kg) and 70 µg/kg (cumulative dosage 100 µg/kg). These changes have been carried out in the revised manuscript. Please see page 5 line 132-133, Figure 3A , page 7 line 147-149 (Figure 3A  legends), page 13 line 371-374.

A critical point relies in the lack of tissue analysis after PAR2 inhibitor. Why was it not done? One could expect some effects. Also, why did authors choose this inhibitor? Are there any MCT inhibitors available that could be used to validate the results obtained?

# Response

Thank you very much for your comment. The tissue changes caused by ischemia are well known to be irreversible. Previously, we demonstrated the tissue changes with ischemia were prevented by administration of Rho-kinase inhibitor or tetrahydrobiopterin for 8 weeks (Akaihata, Sci Rep. 2020. Akaihata, Urology. 2017). In this study, we injected the PAR2 antagonist only 3 times sequently, which should be too short to induce any histological changes by the PAR2 antagonist .

We chose FSLLRY-NH2 as a PAR2 antagonist in this study because Xing J et al injected FSLLRY-NH2 against male Sprague-Dawley rats (Xing J, Cell Physiol Biochem. 2017). We did not try other drugs.

APC 366, which is a selective inhibitor of mast cell tryptase, was reported to attenuate neuroinflammation in rat. APC 366 may be used to validate our results.

The 1st paragraph of the discussion should be divided into sections. There are too many subjects being discussed at the same time.

# Response

Thank you very much for your suggestion. This change has been carried out in the revised manuscript.

The hypothesis of the lack of effect of 30 and 100 µg needs to be better explained. Are there any alternative hypotheses?

# Response

Thank you very much for your suggestion. In this study, our hypothesis was the inverted U-shaped dose-effect curve. We would like to check the effect of varying the dosage in smaller steps, such as 5 µg/kg to 15 µg/kg.

Minor points:

In section 2.1, there is no need to indicate values are they are described in table 1.

In figure 3B, please include some notification (asterisk) showing statistical differences. As in other figures, include that information in the legend, as well.

To the less experienced reader, in the discussion it is advisable to clarify that the myelinated A- fibres are delta fibres (and not alfa or beta).

# Response

Thank you very much for your suggestions. These changes have been carried out in the revised manuscript. Please see page 2 line 84-85, Figure 3.B, and page 10 line 215.

Reviewer 2 Report

In this manuscript the authors attempt to demonstrate a relationship between mast cell infiltration, urothelial barrier dysfunction and bladder hyperactivity following chronic bladder ischemia.  The authors show that 8 weeks following damage to the endothelium of both iliac arteries, the number of mast cells infiltrating the bladder increased, the expression of mast cell tryptase (MCT) and protease-activated receptor 2 (PAR2) increased, and the expression of Uroplakin II decreased.  Additionally, IV treatment of rats with a PAR2 antagonist improved bladder function in cystometry. 

Major Concerns:

1.       My main concern is that much of the data is used to make conclusions that should be confirmed through further experimentation.  For example, the authors conclude that a down-regulation of UPII means that urothelial barrier function is impaired.  It is unclear whether this reduction actually leads to barrier dysfunction, which would have to be measured ex vivo in an Ussing chamber.  Alternatively, the authors could expand their molecular characterization of urothelial barrier function by looking at changes in expression of tight junction proteins (e.g., claudins, ZO-1).

2.       The change in uroplakins is also rather correlative; it would have been better if the authors examined if treatment with the PAR2 antagonist (through use of an osmotic pump or IP injections) reversed the decrease in UPII. 

3.       How does the bladder activity in the CBI model compare to non-injured controls?  It cannot be said that the model causes bladder hyperactivity if it is not compared to non-injured controls.

4.       Eight weeks following injury seems to be a rather long time after injury to see an effect.  In fact, the results the authors report (i.e., mast cell counts, UPII downregulation) appear to be underwhelming.  I wonder if the results would be more significant if the post-CBI time was shortened.

Minor Concerns:

1.       It is not necessary to have both Table 2 and Figure 3B.  One should be removed (I would remove Figure 3B).  Also, statistically significant changes should be noted in whichever one is kept.

2.       The article is well-written, but could still benefit from an additional round of editing from a person who speaks English natively. 

Author Response

Thank you for Reviewers comment. We hope we have made our paper understandable.

Point by point response

Reply to Reviewer 2

Major Concerns:

  1. My main concern is that much of the data is used to make conclusions that should be confirmed through further experimentation. For example, the authors conclude that a down-regulation of UPII means that urothelial barrier function is impaired. It is unclear whether this reduction actually leads to barrier dysfunction, which would have to be measured ex vivo in an Ussing chamber.  Alternatively, the authors could expand their molecular characterization of urothelial barrier function by looking at changes in expression of tight junction proteins (e.g., claudins, ZO-1).

# Response

The reviewer is correct. In this study, we did not confirm that the down-regulation of UP II causes urothelial barrier dysfunction. We considered that this is a limitation. However, some studies demonstrated that UPs were an important part of apical membrane which acted urothelial exceptional barrier (Lavelle, The American journal of physiology. 1998) and that down-regulation of UPs caused loss of urothelial barrier function and MC infiltration (Cetinel S, BJU international. 2011). So, we speculate that decreased UP II expression causes MC infiltration in the bladder through urothelial barrier dysfunction. Please see page 11 line 262-264.

  1. The change in uroplakins is also rather correlative; it would have been better if the authors examined if treatment with the PAR2 antagonist (through use of an osmotic pump or IP injections) reversed the decrease in UPII.

# Response

Thank you very much for your suggestion. As a next step, we would like to examine the effects of the PAR2 antagonist on the decreased UP II expression in the CBI rat model. Our hypothesis is that chronic ischemia caused down-regulation of UP II, MC infiltration and increased PAR2 in the bladder. We consider that the PAR2 antagonist do not reverse the decrease in UP II because the PAR2 antagonist may not improve bladder blood flow.

  1. How does the bladder activity in the CBI model compare to non-injured controls? It cannot be said that the model causes bladder hyperactivity if it is not compared to non-injured controls.

# Response

Thank you very much for your comment. Previous studies including our previous studies demonstrated bladder hyperactivity in the CBI rat model used in the present study showed, defined as shortened micturition interval, decreased bladder capacity and voided volume, without affecting maximum pressure or residual volume compared to non-injured controls (Akaihata, Sci Rep. 2020. Nomiya, J Urol. 2013. Kim JW, J Urol. 2013). In this study, we showed decreased mean voided volume and maximum voided volume in the CBI group as compared with control group in the metabolic cage studies. We consider that the CBI rat model had bladder hyperactivity in this study.

  1. Eight weeks following injury seems to be a rather long time after injury to see an effect. In fact, the results the authors report (i.e., mast cell counts, UPII downregulation) appear to be underwhelming. I wonder if the results would be more significant if the post-CBI time was shortened.

# Response

This CBI rat model was developed to evaluate the effect of ischemia/reperfusion injury on bladder function in our laboratory, in order to disclose the mechanisms involved in the lower urinary tract dysfunction in the elderly patients with atherosclerosis. In our previous studies, we found that it needs 8 weeks to create arterial occlusive disease in bladder microvessels as well as iliac arteries by balloon endothelial injury with hypercholesterolemia (Akaihata, Sci Rep. 2020. Nomiya, Neurourol Urodyn. 2012). So we consider 2% cholesterol diet for 8 weeks play a key role to evaluate lower urinary tract dysfunction by ischemia/reperfusion injury.

Minor Concerns:

  1. It is not necessary to have both Table 2 and Figure 3B. One should be removed (I would remove Figure 3B). Also, statistically significant changes should be noted in whichever one is kept.

# Response

Thank you very much for your suggestion. We removed Table 2 because Figure 3B clearly illustrates the effect of PAR2 antagonist on the bladder function of the CBI rat model.

  1. The article is well-written, but could still benefit from an additional round of editing from a person who speaks English natively.

# Response

We have requested English editing of our revised manuscript by a native speaker.

Round 2

Reviewer 1 Report

Nothing further to point out.

Reviewer 2 Report

The authors have addressed my earlier concerns/comments.